# Concept Evaluation for the Establishment of a Firm End-Stop Feeling in an Asymmetric Hydraulic Steering Unit

**Emil N. Olesen** [1,*], **Torben O. Andersen** [2] **and Henrik C. Pedersen** [2]

1  Danfoss Power Solutions Aps, Grønvej, 6430 Norborg, Denmark
2  Fluid Power Technology, AAU Energy, Pontoppidanstræde 111, 9220 Aalborg, Denmark;
   toa@energy.aau.dk (T.O.A.); hcp@energy.aau.dk (H.C.P.)
*  Correspondence: emil.olesen@danfoss.com

**Abstract:** Danfoss Power Solutions Aps has a product line focusing on hydraulic steering units for heavy-duty machines. The focus of this paper is on the end-stop torque encountered by the operator for a new asymmetrical hydraulic steering unit, referred to as sSteer. This hydraulically asymmetric concept increases the steering responsiveness between the steering wheel input and the output. However, compared to traditional hydraulic steering units, the asymmetrical design has a drawback regarding the level of end-stop torque felt by the operator when reaching the left-side end stop. This paper investigates three different concepts for improving/increasing the end-stop torque, namely, including a bleed orifice, removing a set of suction valves, and a solution with pre-tensioned suction valves and tank line. During the investigations, these concepts were compared and benchmarked using experimental data to identify advantages and disadvantages. Based on the investigations, it is concluded that the concept with pre-tensioned suction valves and a pressurized tank line ensures the best compromise between the different design requirements and the establishment of a firm end-stop feeling for the operator.

**Keywords:** power steering; rotary spool/sleeve set; end-stop torque; asymmetrical hydraulic steering unit; off-road machinery; control

## 1. Introduction

The hydraulic power steering system is a crucial part of off-road vehicles, allowing for the operator to safely and comfortably operate everything from a large-scale tractor in the agricultural field to a material-handling vehicle at a construction site. This type of application variance calls for a flexible and robust product platform that can be easily installed and maintained. Hydraulic power steering systems can, in general, be split into two main categories, which are shown in Figure 1. The left side of Figure 1 illustrates a hydraulic steering system wherein a steering wheel combined with a steering column actuates a hydraulic steering unit. This unit controls the wheel movement depending on the steering wheel input, where it meters flow to either side of the steering cylinder controlling the wheels. The right side of Figure 1 illustrates an electrohydraulic system, where a given input device controls the wheel movement through an electrohydraulic steering unit, which also meters flow to either side of the cylinder depending on the input.

Productivity is a key motivation for the continuous improvement of off-road vehicles; increased speed limit regulations in various European countries [1] are among the improvements that significantly affect the day-to-day efficiency of operators. A disadvantage of a higher vehicle speed is that the mental workload of the operator during steering is increased due to increased vibrations in the vehicle [2,3]. It is, therefore, of interest to ease the steering of such vehicles. The operability or ease of steering an off-road vehicle, as illustrated in Figure 2, highly relies on the translation between the desired steering trajectory from the operator and the ability of the power steering system to transfer this trajectory into a wheel movement (output) through the steering wheel or input device (input).

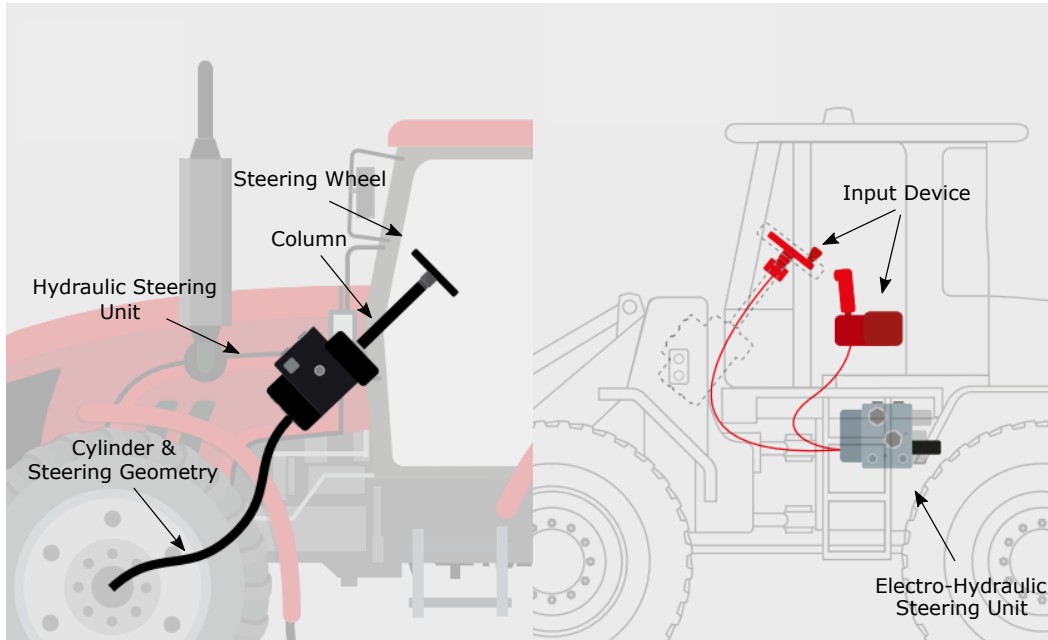

**Figure 1. LEFT** Illustration of a hydraulic steering system implemented on a tractor. The steering system comprises a steering wheel, steering column, steering unit, and the vehicle steering geometry. **RIGHT** Illustration of an electrohydraulic steering system implemented on a wheel loader. The system consists of a steering wheel, joystick, electrohydraulic steering unit, and the vehicle steering geometry.

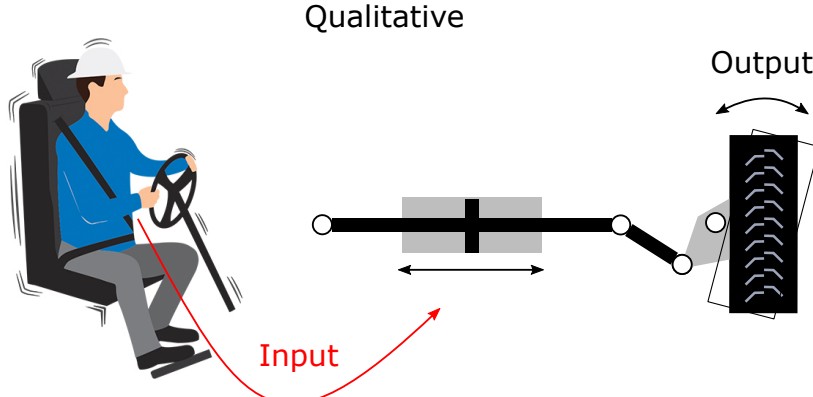

**Figure 2.** Illustration of the operability term for steering systems, which relies on the translation between the operator input to the steering system and the wheel angle of the vehicle.

The response of the steering system can be measured with a number of sinusoidal steering inputs, which can then be related to a diagram or transfer function for the closed-loop system. The input is the steering wheel angle, and the output is the cylinder position. However, the operability or ease of using a steering system cannot be quantitatively evaluated. Instead, this is qualitatively measured by experienced test operators, influenced by parameters such as steering torque, steering wheel angle versus wheel, how aggressive or passive the gain of the steering system is, and the dead band of the steering system [4], compared to the vibrations of the vehicle, cabin or steering wheel [5]. Therefore, the operability depends on the operator as part of a closed-loop system, which makes it more complex to determine the input and output for the system. The output is ideally the desired vehicle direction combined with a comfortable feeling of the steering system for the operator, while the input is the operator's interaction with the steering wheel or input device.

A new state-of-the-art steering concept, developed by DPS, called sSteer increases the operability of the vehicle [6,7]. This concept is based on an asymmetrical hydraulic designed steering unit. Figure 3 shows the ISO diagrams for a conventional steering unit and the new asymmetric design. In a conventional unit, the steering oil flows from the P-line, through the mechanical actuated linear directional valve, through the metering motor to one of the cylinder sides. Simultaneously, the return oil from the cylinder is led through the directional valve to the tank. In the new concept, the flow path is changed, such that, when steering to the right, the steering oil flows through the directional valve and the metering motor, and enters the right side of the cylinder while the return oil is led through the directional valve and to the tank. When steering to the left, the steering oil flows through the directional valve and directly enters the left side of the cylinder, while the return oil from the cylinder flows through the metering motor and then to the tank through the directional valve. The difference in the diagram allows for the designer to create an under-lapping directional valve for the asymmetric unit, which is desirable as it enables pressure feedback in the mechanical steering wheel when the underlapping area is in operation. In this area, it is possible to obtain the relation between the pressure balancing of the steering cylinder and the mechanical torque of the steering wheel felt by the operator. This torque versus pressure balance relation can be compared to driving a car, where the operator can feel the reaction forces on the tire when turning the steering wheel. Secondly, the pressure balancing principle will self-align the vehicle's tires when the steering wheel is let go, as in a car. The underlapping design and hydraulic asymmetry can be placed in either side of the unit; in the following description, it is placed in the right side.

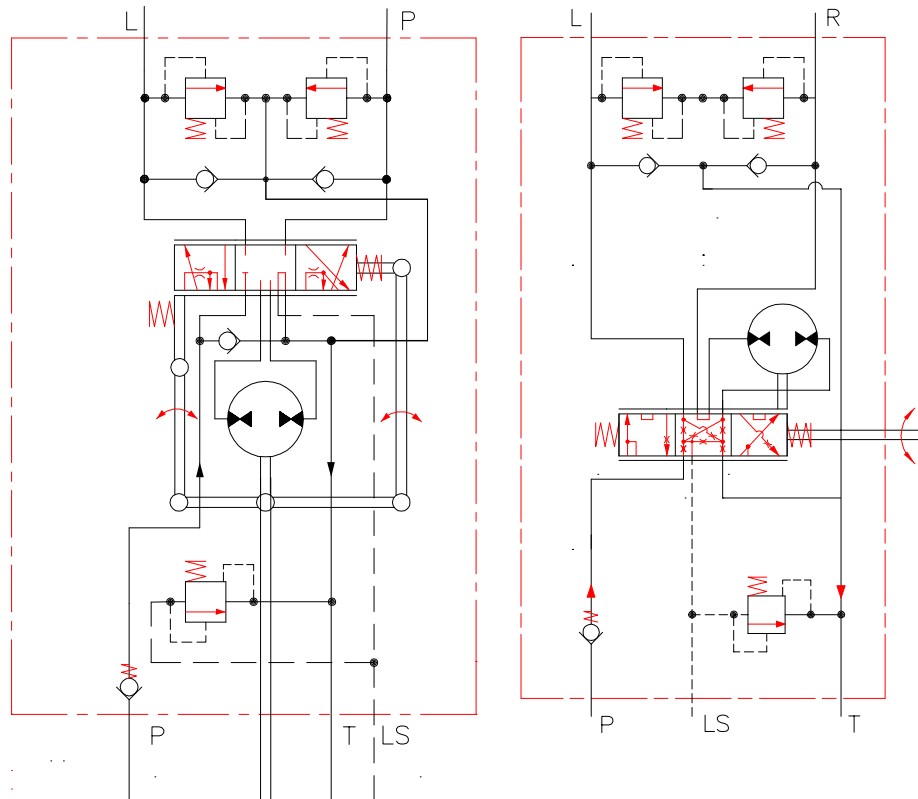

**Figure 3.** In general, for both diagrams, a hydraulic steering unit is shown, which consists of a P-check valve, a directional valve, a metering motor, suction valves, and pressure relief valves. **LEFT** A ISO hydraulic diagram for a conventional OSPC steering unit. **RIGHT** A ISO hydraulic diagram for an asymmetric OSPS steering unit.

However, this automotive steering feeling comes with some disadvantages; one is a significant decrease in the left end-stop torque when the steering cylinder reaches the end

stroke and the relief valve is activated. In this situation, a conventional steering unit applies a symmetrical torque for both left- and right end-stop from 10 Nm and above, depending on the metering motor size. In the new sSteer concept, the end-stop torque is decreased to only a few Nms to the left. The right-side end-stop of the sSteer remains a conventional unit. It is, therefore, not possible for the operator to notice the left end-stop situation if the end-stop torque to the left is not improved. Instead, the operator will continue to rotate the steering wheel in the left end-stop without knowing that the vehicle cannot turn any more sharply.

A general drawback for hydraulic and electrohydraulic valves is the significant low-energy efficiency, which, according to [8], results in average system efficiencies of around 22% when considering the entire hydraulic industry in the US. This phenomenon is also a topic for steering units due to the throttle losses in the directional valve, which directs the oil to and from the steering cylinder. The disadvantage is an inefficient system, but the advantages are an excellent dynamic response, good durability, and a low noise level in the cabin of the steering units. It is, therefore, not straightforward to increase the energy efficiency of a steering system without compromising its performance. Noise will be an especially hot topic in the future if electrification completely replaces engine noise. According to the studies [9,10], the energy efficiency of the steering system can be improved by rearranging the hydraulic actuation system. In a conventional steering system, a hydraulic or electrohydraulic unit is supplied through a pump, which ensures a certain margin pressure. This setup will consume an amount of energy even when the steering unit is not activated due to the margin pressure level. In the aforementioned studies, the steering unit is removed from the setup and, instead, a pump is added, which is directly connected to the steering cylinder(s). This pump is then volumetrically controlled through an electric motor, such that an input signal from a steering wheel or joystick is directly translated into a metered flow and, thereby, a cylinder movement occurs as output. According to the studies, a wheel loader application can save up to 50% of the energy usage during steering, mainly by removing the throttle losses from the conventional system. On the other hand, the number of components introduced with this kind of system is significantly higher, which therefore affects the return of investment (ROI) of the vehicle. In addition, the throttle losses in a conventional system are designed to dampen the steering system. Therefore, it would be interesting to see how the pump concept works in an experimental vehicle setup.

As the end-stop feeling is a crucial parameter for a steering system, it is necessary to increase the left-side torque level for sSteer to obtain an acceptable performance for the customers. In a recent study [11], an end-stop concept is investigated. The study concluded that the left end-stop torque could be increased, but the given solution had the drawback of introducing undesired steering wheel oscillations when the left end-stop was reached. These oscillations annoy the operator during steering and are, therefore, not acceptable.

The objective of this paper is to analyze the end-stop situation, where the focus will be on investigating three different concepts to increase the end-stop torque combined with an experimental verification of the concepts. The paper will not focus on a detailed modelling of the end-stop situation. This has previously been studied in combination with a sensitivity study in the paper [11]. The new contributions, compared to [11], are the evaluation of two new concepts and a general comparison of the earlier-studied concept combined with these. The paper will consist of the following parts:

- Endstop Concept Analysis;
- Experimental Setup;
- Experimental Results;
- Discussion and Conclusion.

The first section will describe and analyze three different concepts developed to achieve the desired torque level for the left side. The second section will specify the experimental setup. The third section will describe how the concepts are tested and show the corresponding results. Finally, the results and outcome of the experimental tests and

analysis of the concepts are discussed, and the best-performing concept is identified as a conclusion.

## 2. End-Stop Concept Analysis

To understand the problems facing the sSteer concept, an exploded view of a sSteer unit is shown in Figure 4, together with a hydraulic diagram of the unit. The sSteer concept consists of the following orifices, A1, A2L, A2R, A3L, A3R, and A10, which are integrated into the spool and sleeve, a gearset (marked with green), referred to erlier as a metering motor, suction valves (marked with blue), and a relief valve (marked with grey), which ensures that the maximum load pressure of the unit is not exceeded.

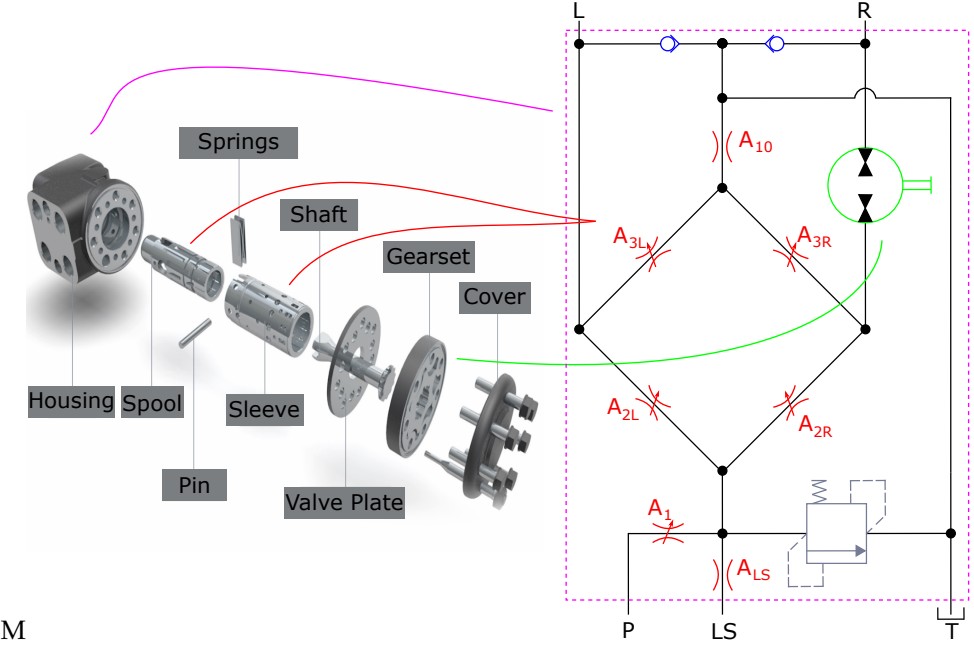

**Figure 4. LEFT** An exploded view of the steering unit with the corresponding components. **RIGHT** A hydraulic diagram for the sSteer concept.

The sSteer unit functions as a closed-loop control system with mechanical feedback. The input is the angular movement of the spool, which is linked to the steering wheel through the column. The output is the movement of the steering cylinder/wheels of the vehicle, which is linked to the angular movement of the sleeve. In the steering unit, the spool and sleeve are connected with a spring package, which creates the mechanical feedback. The angular movement of the sleeve and the steering cylinder and, hence, wheel angle movement depends on a hydraulic gearing ratio, which is determined by the size of the gear set. This analysis will focus on the end-stop situation. A more general description of the sSteer concept can be found in the paper [7].

The sSteer concept is asymmetrical due to the position of the gearset, which, in a conventional unit, is placed symmetrically between the A2 and A3 bleed rows. Figure 5 visualizes the end-stop situations for the left and right sides, respectively. At the end-stop, the gearset is mechanically linked to the steering wheel, and therefore, it is possible for the operator to rotate the gearset by applying a torque.

When the operator reaches the end-stop in the right side, the oil flow will increase the pressure towards the relief setting due to the changing cylinder volume, which becomes constant. The volume becomes constant due to the mechanical endstroke of the steering cylinder. At the end-stop, there will be an equal pressure acting on both sides of the gearset, which will provide a firm end-stop feeling for the operator that depends on the gearset size and the slippage level of the gearset. This torque is, in general, above 10 Nm for

conventional steering units, which indicates to the operator that the end-stop of the steering geometry for the vehicle is reached.

When the operator reaches the end-stop in the left side, the oil flow will again increase the pressure towards the relief setting due to the changing volume. However, it will not build up an equal pressure on the gearset as for the right-side situation; instead, it will act as a pump. The pumping will occur when the operator applies a torque on the steering wheel, which will generate a flow following the path illustrated in Figure 5 with blue arrows. The inlet flow is sucked through the suction valve, which is indicated with the green arrow in Figure 5. The endstop torque for the left side will, therefore, only depend on the friction in the gearset, which, for a conventional gearset, is below 0.5 Nm. The felt endstop is thus significantly decreased for the left side, which is not accepted as a clear indication of the endstop.

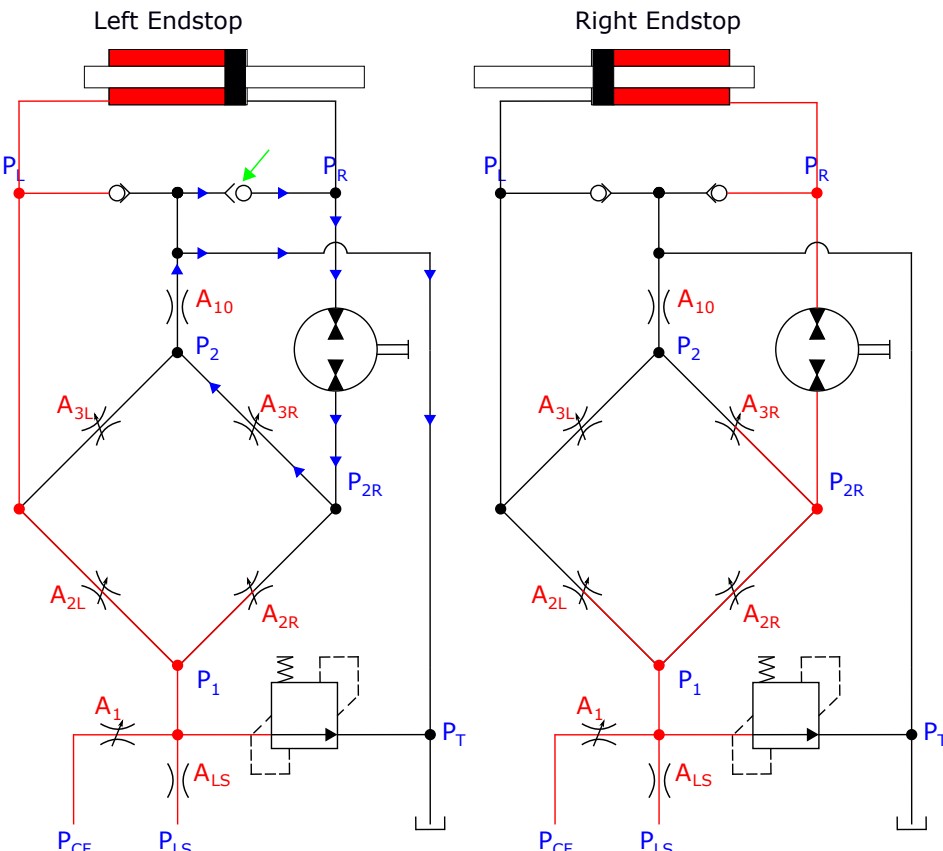

**Figure 5.** **LEFT** A hydraulic diagram for the left end-stop situation. The red line indicates the high-pressure side, and the black lines are connected to the tank line. **RIGHT** A hydraulic diagram for the right end-stop situation.

To improve the end-stop torque at the left side, it is thus necessary to affect the gearset rotation. This can be limited in different ways Figures 6 and 7 show three different concepts investigated to do so.

Concept (a) consists of an extra bleed row, called A1314, which is able to increase the pressure point $P_{2R}$ by opening up for a leakage path between the LS line and the A10 bleed when the end-stop is reached. The increased pressure level of $P_{2R}$ will increase the loading of the pump, and it will therefore require a higher torque to rotate the steering wheel.

Concept (b) removes the suction valves. This change restricts the inlet oil for the gearset pump to be sucked from the tank side. The torque level will, therefore, be increased because it is necessary to fill the inlet with leakage oil instead. This design change can introduce cavitation of the metering motor, specifically point $P_R$ on the figure, but some

customers do not utilize suction valves because they are not a requirement for the general market, and it is therefore of interest to see the effect of this concept.

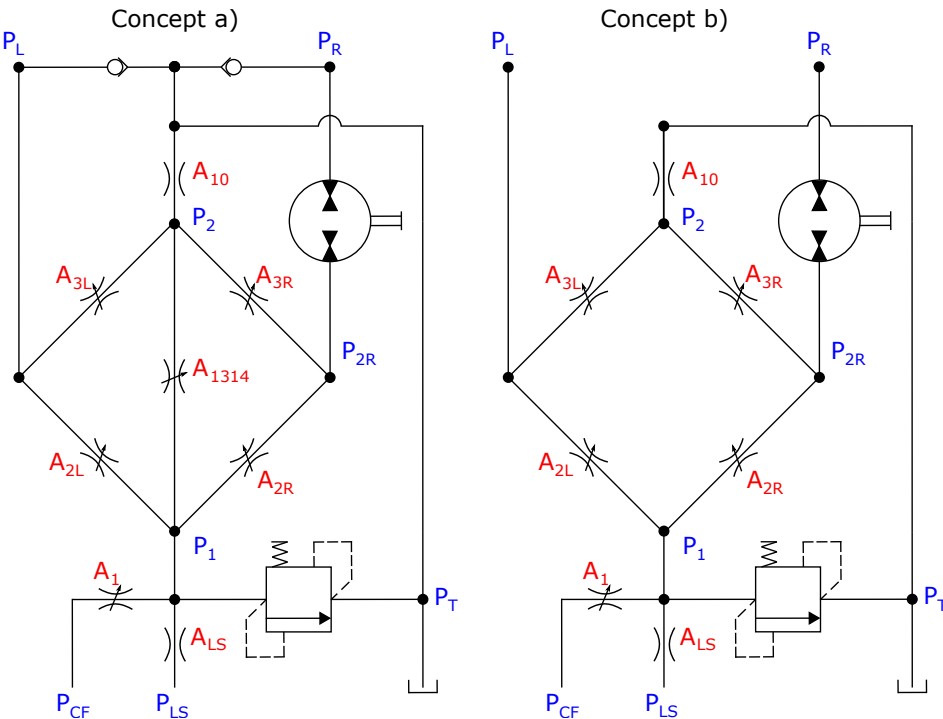

**Figure 6.** (a) A hydraulic diagram for the end-stop concept with an extra bleed row, A1314. (b) A hydraulic diagram for the endstop concept without suction valves.

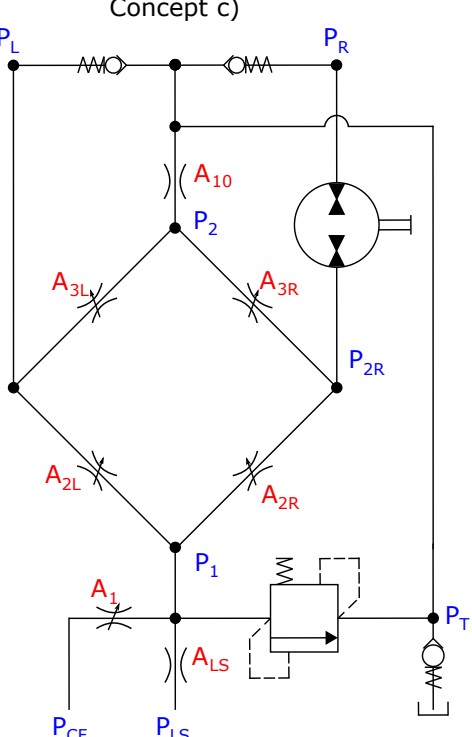

**Figure 7.** (c) A Hydraulic diagram for the endstop concept with pretensioned suction valves and tank line.

Concept (c) consists of pretensioned suction valves and a pretensioned tank check valve. This concept raises the entire tank pressure level of the steering unit to achieve an end-stop torque of 10 Nm, with from 3.5 to 8 Bar, depending on the gearset size. The end-stop torque is increased due to the pretensioned suction valves, where it is necessary for the operator to apply a high enough torque to start opening the suction valves.

The endstop torque for concepts (a) and (c) can be calculated using the linear relationship between torque, displacement, and pressure for a hydraulic pump. This is shown in Figure 8 with 10 Nm as the desired endstop torque. Regarding concept (b), the end-stop depends on the slippage of the given gear set, which will vary depending on the production clearance. It is, therefore, difficult to predict the end-stop torque without verifying it experimentally.

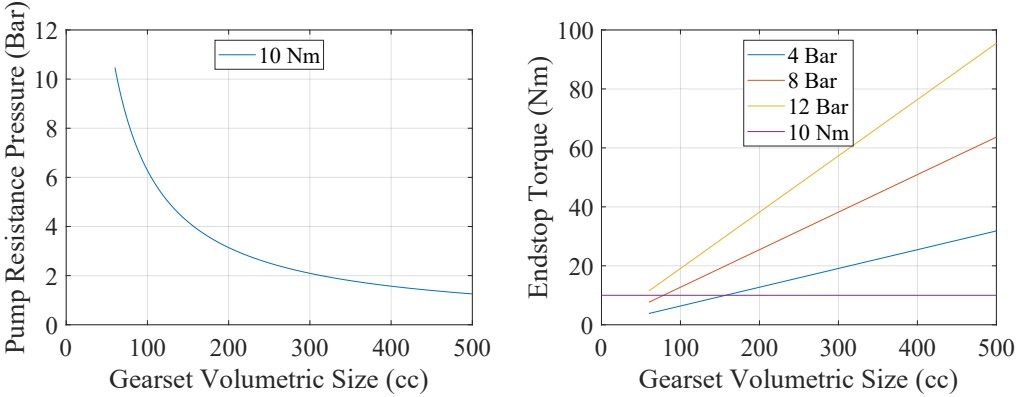

**Figure 8. LEFT** A graph of the delta pressure necessary to generate a start torque of 10 Nm. **RIGHT** A graph of the endstop torque depending on the pretensioned suction valves—4 Bar, 8 Bar, and 12 Bar—for a range of gearset sizes.

The three concepts are built into a prototype unit, which is tested in a laboratory environment and a tractor at Danfoss Power Solutions plant in Nordborg, Denmark. Concept (a) is experimentally verified in the paper [11], and the same experimental data and results are shown again to provide a full overview.

## 3. Experimental Setup

The experimental verification of the concepts is first performed in a controlled laboratory environment with constant oil temperature and calibrated measurement equipment. The accuracy, oil properties, and the setup are described in the following. Secondly, the steering unit(s) is tested at an application and development center on a commercial tractor, which will briefly be described as well.

The experimental tests for concepts (b) and (c) are performed with a sSteer Load Sensing steering unit, which has a gearset size of 305 cm$^3$, while concept (a) is performed with a gearset size of 125 cm$^3$. For concept (b), the suction valve bores are blocked in the housing, and for concept (c), the suction valves are pre-tensioned with 3.5 Bar that provides a theoretical endstop torque of 17 Nm. For concept (c), the tank line is also pretensioned to 4–5 bar.

The important evaluation criteria in an end-stop test are, first of all, the level and oscillations of the torque delivered in the left and right end-stop situation. Secondly, the level of slippage of the metering motor when this torque is applied, and finally, the amount of tank flow or total leakage in this end-stop position.

### 3.1. Laboratory Setup

The test bench is shown in Figure 9 to the left, which consists of a fixed table, where a steering wheel and column are attached on top. Under the table, an electric motor and a planetary gear are fixed, which is able to operate the steering wheel through a belt transmission. At the other end of the column, it is possible to mount a given steering

unit. The cabinet on the left side of the table is integrated with a hydraulic supply unit. The supply unit is a gear pump whose input flow can be controlled. This gear pump is connected to a priority valve, which is shown in Figure 9 in the hydraulic diagram to the right. The priority valve is connected to the steering unit, where the priority valve is able to control the amount of CF flow for the steering unit based on the LS signal. The remaining pump flow is let through the EF channel towards the tank. The servo side of the steering unit (R & L) can be connected to either a cylinder, a controlled hydraulic resistance, short-circuited or blocked. The test carried out for this paper is used blocked ports to simulate an end-stop situation. From Figure 9, pressure (indicated with blue), flows (indicated with green), torque, angle, and velocity (indicated with red) points are measured and logged with a frequency of 100 Hz. The test frequency is relatively low, but the test is not performed to catch dynamic responses.

- Hydraulic Oil—Shell Tellus S2 MX 32
- Inlet pump flow—80 L/min
- Margin pressure of priority valve (CF-LS)—12 bar
- Inlet flow from LS—1.3 L/min
- Oil Temperature—50 degrees Celsius
- Calibrated pressure sensors with an accuracy of +− 1 bar
- Calibrated torque sensor with an accuracy of +− 0.01 Nm
- Calibrated speed and angle sensor with an accuracy of <0.1 degree
- Calibrated flow sensor +− 0.1 L/min

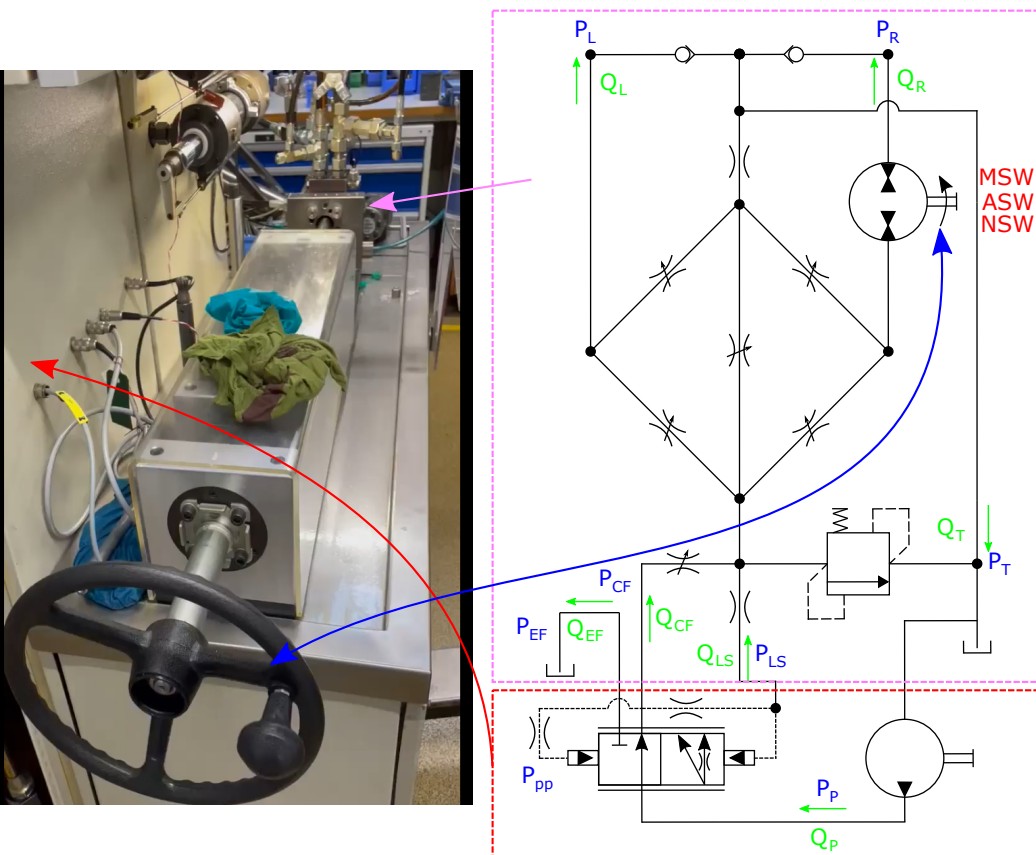

**Figure 9. LEFT** A picture of the laboratory setup with the steering column, steering unit, and measurement equipment. **RIGHT** A hydraulic diagram of the laboratory setup for the specific end-stop evaluation. Pressure points measured are indicated in blue, flows in green, angular movement and torque in red. $Q_i$ is measured flow lines, $P_i$ is measured pressure points, MSW is the measured torque, NSW is the measured velocity, and ASW is the measured angle.

### *3.2. Tractor Setup*

The experimental tractor is a 350-horsepower medium-sized tractor without any implements. The hydraulic system consists of a variable displacement LS pump for the entire vehicle, where the steering system is supplied through a priority valve, which is directly mounted on the pump. The priority valve has a margin pressure between CF-LS of 12 bar and an LS inlet flow of 1.3 L/min. The measurement equipment is logged for the laboratory setup, with a frequency of 500 Hz. The equipment has the following properties:

- Oil Temperature—60 degrees Celsius;
- Calibrated pressure sensors with accuracy of +− 1 bar—Range of 0–600 Bar;
- Calibrated torque sensor with an accuracy of +− 0.01 Nm;
- Calibrated speed and angle sensor with an accuracy of <0.1 degree;
- Calibrated flow sensor +− 0.1 L/min.

## 4. Experimental Results

The experimental results will be split into three parts, corresponding to the number of concepts, where the results for both the laboratory test and the tractor test will be discussed and shown.

### *4.1. Concept (a)*

The lab test was performed with a manual operator input, and two different tests were performed. The test results are shown in Figures 10 and 11. On the left, the figures show the applied end-stop torque and corresponding steering wheel velocity depending on time. On the right, the figure plots the tank line flow from the steering unit to indicate when the A1314 orifice is activated.

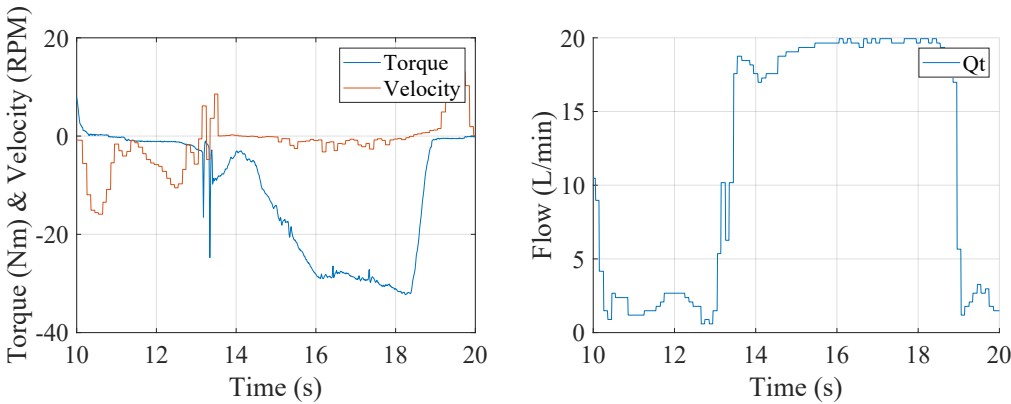

**Figure 10. LEFT** A graph of the end-stop torque and steering wheel velocity for a manual operator input. **RIGHT** A graph of the measured tank flow for the manual end-stop input. Both graphs show data from a prototype with the concept (a) configuration.

The results from the first test, shown in Figure 10, show that the A1314 orifice is activated at around 13 s due to the increase in tank flow. At this time, the torque level is increased to a level of around 10 Nm, and the steering wheel velocity is significantly decreased. From these results, it is clear that when the A1314 orifice opens, the torque level is increased to a level of 10 Nm or above, but there is some initial variation.

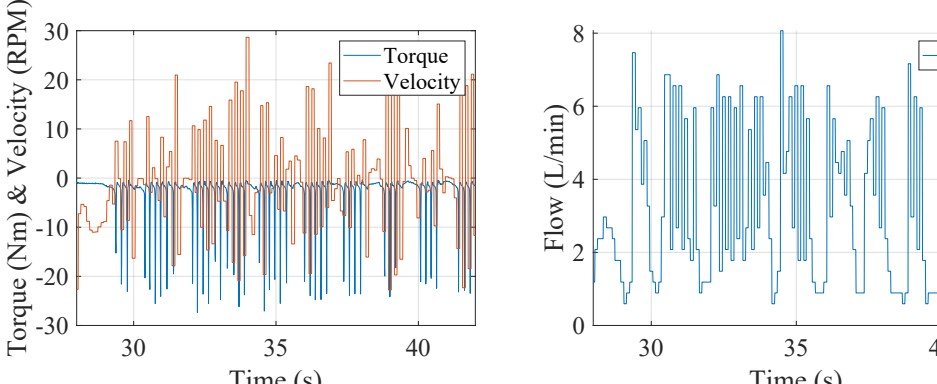

**Figure 11. LEFT** A graph of the end-stop torque and steering wheel velocity for a manual operator input. **RIGHT** A graph of the measured tank flow for the manual end-stop input. Both graphs show data from a prototype with the concept (a) configuration.

The results from the second test, shown in Figure 11, show that the torque and steering wheel velocity are oscillating. This oscillating phenomenon is introduced due to the opening and closing of the A1314 orifice [11]. The opening and closing of the A1314 orifice are indicated on the right graph, in Figure 11, due to the oscillating tank flow. During manual steering, the rate of torque and level have an influence on this phenomenon. If an operator applies a step response of constant torque level—5 Nm and above—the steering wheel will not oscillate. If a lower and inconstant torque level is applied, the oscillations will, however, be provoked due to the dynamics of the system, as also discussed in [11].

From the experimental results, there are two drawbacks to the concept. The first drawback is shown in Figure 10 from the time stamp 10–13 s, where the A1314 orifice is closed, and the end-stop torque is below 10 Nm. In this period of time, it is possible to turn the steering wheel with a velocity of up to 15 rounds per minute without feeling a significant end-stop torque. This phenomenon is called steer-through and is caused by the deflection torque from the neutral positioning springs. The A1314 orifice is first opened when the neutral springs are almost in maximum deflection to ensure that no leakage paths are introduced during regular steering. However, this applies torque on the gearset before maximum deflection, which will cause a pumping mode without any loading. The phenomena can be avoided by introducing a small amount of friction in the gearset, such that the spring package needs to be fully deflected before the gearset can be rotated. The second drawback is the steering wheel oscillations, whose occurrence is described and analyzed in detail in the paper [11], but constitute unacceptable behavior for the system.

*4.2. Concept (b)*

The lab test is performed with a ramp input, where the velocity of the steering unit is increased from 0 to 20 RPM while the end-stop torque is measured. This is shown in Figure 12, to the left.

The tractor test is performed with a manual operator input at the end-stop, where an increasing velocity input is shown in Figure 12, to the right. Here, the end-stop torque $M$, steering wheel velocity $v$, and steering cylinder movement $Xcyl$ are measured depending on time.

The laboratory results show that the end-stop torque is oscillating around 15 Nm, where the amplitude of the oscillation is slowly increasing from $\pm2$ Nm to $\pm5$ Nm.

The tractor results show the same tendencies, where the end-stop torque is again oscillating around 15 Nm. However, the oscillations are decreased in amplitude, which could be due to an increased volume and the flexibility of the hoses in the setup.

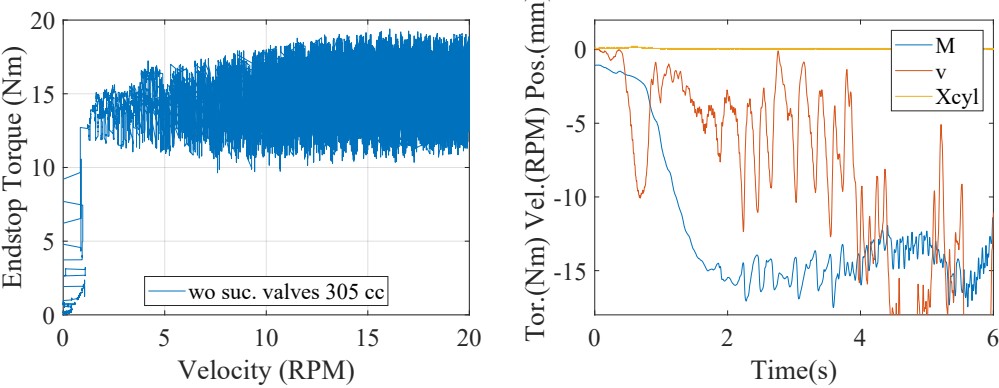

**Figure 12. LEFT** A graph of the delta pressure necessary to generate a start torque of 10 Nm. **RIGHT** A graph of the endstop torque depending on the pretensioned suction valves, 4 Bar, 8 Bar, and 12 Bar, for a range of gearset sizes. Both graphs show data from a prototype with concept (b) configuration.

The torque oscillations are further investigated in the laboratory, where it is shown that the oscillations come from cavitation of the gearset. This is shown in Figure 13, where the left side indicates an end-stop test with suction valves, and the right side indicates a test without suction valves. The test without suction valves introduces air bubbles in the oil, which indicates that the gear set is cavitating. The investigated oil is coming from the tank line from the steering unit.

### With Suction Valves　　　　　　　Without Suction Valves

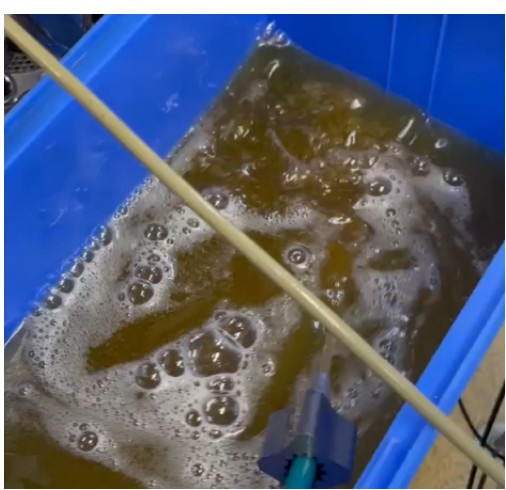

**Figure 13. LEFT** A picture of the tank oil from the steering unit in the end-stop, with suction valves. **RIGHT** A picture of the tank oil from the steering unit in the end-stop, where the suction valves are removed.

*4.3. Concept (c)*

The lab test is performed with a ramp input, where the velocity of the steering unit is increased from 0 to 20 RPM while the end-stop torque is measured. This is shown in Figure 14, to the left.

The tractor test is performed with a manual operator input at the end-stop, where an increasing velocity input is shown in Figure 14, to the right. Here, the end-stop torque, steering wheel velocity, and steering cylinder movement are measured depending on time.

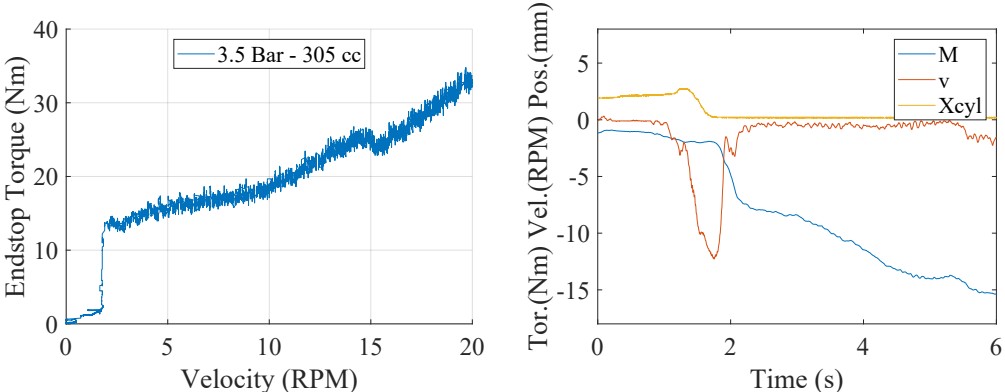

**Figure 14. LEFT** A graph of the delta pressure necessary to generate a start torque of 10 Nm. **RIGHT** A graph of the endstop torque depending on the pretensioned suction valves, 4 Bar, 8 Bar, and 12 Bar, for a range of gearset sizes. Both graphs show data from a prototype with concept (c) configuration.

The laboratory results show that the end-stop torque is starting at 14–15 Nm with an increasing path up to 32 Nm at 20 Rpm. From the offset point of 14–15 Nm, the tendency is linear and smooth with small torque amplitude oscillations of around 0.5 Nm.

The tractor results show the same tendencies, with an offset torque of 8–9 Nm and then a linear increasing torque towards 15 Nm, where the steering wheel starts to slip off. This is indicated at the steering wheel velocity curve *v*, which starts to increase at around 5 s.

From the tractor test, there was a difference in the end-stop feeling for concepts (b) and (c), where concept (c) applied a more smooth and stable torque resistance to the steering wheel compared to concept (b), where torque ripples could be felt by the operator.

## 5. Discussion and Conclusion

Now that the experimental results have been presented, it is of interest to compare and discuss the outcome to identify the most promising concept. In the following, the advantages and disadvantages of the three concepts are listed and compared. For all three concepts, the first bullet points refer to the advantages, and the second row of bullet points refers to the disadvantages.

*Comparison of Advantages and disadvantages*

Concept (a):

- **Advantages**
- Able to increase end-stop torque to <10 Nm;
- Flexible design depending on A1314 and A10;
- Low cost (integration in existing component).
- **Disadvantages**:
- Steering wheel oscillations;
- Steer-through before A1314 opens.

Concept (b):

- **Advantages**:
- Gearset size above 200cc can increase end-stop torque to <10 Nm;
- Low cost (removal of components).
- **Disadvantages**:
- Cavitation of gearset;
- Varying end-stop torque depending on gearset size and temperature;
- No suction valve functionality for steering system.

Concept (c):

- **Advantages:**

- Able to increase end-stop torque to <10 Nm;
- Flexible design depending on spring pre-tension.
- **Disadvantages**:
- Higher cost (T-check-valve and pre-tension springs);
- Increased energy consumption.

From the results, it is clear that both Concept (a) and (c) are able to increase the torque level to 10 Nm for all tested gearset sizes, while concept (b) is limited to gearset sizes above 200cc. The concepts (a) and (c) are, therefore, more flexible to variance compared to (b). Besides this, it is shown in the test that, for concept (b), without suction valves, the gearset is cavitating. This phenomenon is well-known within the hydraulic industry to cause noise, vibration, and lifetime issues in hydraulic pumps. It is, therefore, not desirable to further this concept.

A drawback for concept (a) compared to (c) is the introduction of underdamped steering wheel oscillations. These oscillations are investigated in more depth in the paper [11]. Secondly, concept (a) depends on the A1314 orifice to open before an end-stop torque is felt, which, on the small gearset sizes, introduces steer-through phenomena, which comprise a steering wheel slip of up to 10 RPM. The explanation for this phenomenon is due to the spring package, which can generate up to 5 Nm before they are fully deflected (when the end-stop is reached and A1314 is opened). However, this phenomenon can be resolved by introducing a small friction torque on the gearset, which can be achieved by using a sealing o-ring inside the gearset.

A drawback for concept (c) compared to (a) is the introduction of a pre-tensioned tank line, which affects the metered flow through the steering unit with an additional pressure loss equal to the given pre-tension of the suction valves. The gearset size is linked to a group of applications, where small vehicles, trucks, turfcares, small tractors, etc., in general, have a range of from 60 to 200 $cm^3$ gearsets, while larger vehicles, tractors (mid- to high-end), combine harvesters, dumpers, wheel loaders, etc., have a range of from 200 to 520 $cm^3$ gearsets. First of all, there is the change in gearset size, which has an influence on the pre-tension level, but a second parameter is the pressure relief setting. This relief setting is often lower, from 140 to 180 Bar for small vehicles, due to the weight-force ratio of the vehicle's steering geometry. For larger vehicles, this value is often from 180 to 220 Bar instead. The problem with the concept (c) for the small vehicles is that the system pressure level is decreased at the same time as the tank level is raised, such that the percentage influence on the system's energy consumption will be higher compared to the large-scale vehicles. Concept (a) also utilizes a leakage flow, and therefore energy, to increase the torque level, but this energy is only consumed in the end-stop situation, while in concept (c), energy loss occurs at all times of steering. It could, therefore, be more efficient to use concept (a) compared to concept (c) for smaller vehicles due to the energy consumption as well as the fact that the end-stop situation is not activated often for regular steering.

To summarize the discussed advantages and disadvantages for the different concepts, it can be concluded that concept (b) is not suitable for improving the end-stop torque. This is mainly based on the cavitation phenomena, but also on the influence that temperature, clearance, and gearset size have on the torque level, as felt by the operator. For concept (a) and concept (c), it can be concluded that concept (c) is more beneficial due to the fact that concept (a) introduces steering wheel oscillations and steer-through, which is not acceptable. It can, therefore, be concluded that concept (c) is the most suitable concept for improving the left-side end-stop torque level, with the drawback of an increased energy consumption compared to a conventional unit.

**Author Contributions:** Conceptualization, E.N.O.; methodology, E.N.O.; validation, E.N.O., H.C.P. and T.O.A.; formal analysis, E.N.O.; investigation, E.N.O.; resources, E.N.O.; data curation, E.N.O.; writing—original draft preparation, E.N.O.; writing—review and editing, E.N.O., H.C.P. and T.O.A.; visualization, E.N.O.; supervision, H.C.P. and T.O.A.; project administration, E.N.O., H.C.P. and T.O.A.; funding acquisition, E.N.O. All authors have read and agreed to the published version of the manuscript.

**Funding:** This research received external funding from the danish organisation: Innovationsfonden.

**Acknowledgments:** The authors acknowledge Danfoss Power Solutions for support with prototype building, lab facilities and experimental test setup for a commercial tractor.

**Conflicts of Interest:** The authors declare no conflict of interest.

## Abbreviations

The following abbreviations are used in this manuscript:

| | |
|---|---|
| DPS | Danfoss Power Solutions |
| OSPC | Orbital Steering Pump Type |
| OSPS | Orbital Steering Pump Type |
| MSW | Torque measured at the steering wheel |
| NSW | Angular velocity measured at the steering wheel |
| ASW | Angular position measured at the steering wheel |
| CF | Controlled Flow for steering unit |
| EF | Excess Flow to working hydraulics from steering unit |
| LS | Load Sensing signal for steering unit |

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
