# Peer review of "Concept Evaluation for the Establishment of a Firm End-Stop Feeling in an Asymmetric Hydraulic Steering Unit"

_actuators, doi:10.3390/act12110420_

Round 1

Reviewer 1 Report

Comments and Suggestions for Authors

GENERAL

The paper concerns with the experimental work of a new concept of hydraulic steering unit. Three configurations are proposed and studied to improve the end-stop feeling of this new asymmetric concept.

The paper has a length of 12 single-spaced pages including 12 figures, no tables and no equations. I would like to remind the authors to check the instructions and standards of the journal. For instance, sections, captions, references, among others, through the text does not seem to follow the guidelines of the journal.

In general, in the opinion of this reviewer, the paper is a standard well-written. However, the topic could be of interest and the argument is plausible. Authors should complete a better explanation of some aspects of the paper that will be exposed in the next lines.

IMPORTANT COMMENTS

The reviewer has also to admit that this review of the present paper is not straightforward. Firstly, the complexity of a hydraulic steering unit, and even more when a new concept of it is described. As a consequence, the work presented, although well tackled, is quite difficult to be verified by a single peer-review. Finally, the work refers to other researchers and authors’ own works, and it would be necessary to read them in order to reach a deeply understanding when you are not an expert in this particular field.

The reviewer suggests that the abstract would include the main conclusions obtained in this work pointing out the breakthroughs of their work. The abstract should be concise and a good summary of the paper. The last sentence should be the main breakthroughs of their work.

The new state of the art steering concept aSteer should be more in detail explained, but in contrast, comparing with conventional one from also Danfoss. 

For instance, compare figure P301 482 from (https://www.grouphes.com/wc/hes/Danfoss/SteeringUnits/OSPU.pdf) and figures 2 and 4 in the previous authors’ work reference [8]: https://vbn.aau.dk/ws/portalfiles/portal/533724284/A2.2_Emil_N_rreg_rd_Olesen_Analysis_of_End_Stop_Oscillations_in_an_Asymmetric_Hydraulic_Steering_Unit.pdf

As a non-expert in and less instructed in the hydraulic steering unit, I would include the description of the orbitrol and priority valve to compare and contrast to Steer concept in Figure 2. That would help the reader, for instance such as figure 4 in the previous authors’work https://vbn.aau.dk/en/publications/investigation-of-a-new-orbital-steering-concept-with-focus-on-the

It seems that this work comes from a current PhD, and two other symposiums/congress papers related to this work were presented. Then, I recommend improving the description of the new contribution in reference to the previous published works.

The number of references used is quite short. The subject of the paper, as the authors point out, is a long-time line of investigation, and I think that including other references will help the description of the conventional to the cutting-edge hydraulic steering units. Hence, I recommend to extend it. (2 over 8 are from the authors). The reviewer believes this introduction section includes specific State of Art in order to place author’s work and select, comment and refer the most significant ones to ensure that the authors are aware of similar research work performed by other researchers. 

For this reviewer, it is not clear the advantage of the asymmetric. With this configuration, for instance, other elements such a torque compensator valve can be saved? Why the asymmetric has to be in the left end-stop?

Another important question is the details of the instrumentation used in the experimental work have to be provided: brand, accuracy, calibrated span, full scale, resolution, repeatability, response time, etc. For instance, torque measurement response time in figure 10 seems of high frequency.

OTHER QUESTIONS

The new concept presented, as a load sensing steering unit, it still has any type of amplification?

What about the adjective for steering units: hydrostatic or hydraulic? Are interchangeable? 

Since a drawback of the aSteer concept is the energy efficiency compare with conventional steering, are here removing throttle losses?

What about the analytical model, such a 1D lumped parameters, for instance, as presented in previous works but with the three concepts?

The purpose of the suction valves is to allow oil suction to avoid cavitation in the steering cylinder. Then, the concept (b) without them, the result was expected, figure 11.

In addition, it seems that to provide correct suction, a back pressure valve must be fitted in the tank line from the steering unit. This back pressure valve is the one drawn in Figure 5, at the right-down corner? Then, the pretensioned suction valves are not in series with this back pressure valve; in order words, repeating the same function? Please comment it.

FIGURES’ NOTES

-        The arrows of figure 2 should clearly point to the item in the exploded view to its counterpart in the hydraulic diagram.

-        What is exactly the function of the relief valve (marked in grey) in figure 2?

-        The subindexes in the figures must be described, such as ‘CF’ (towards steering??) in figure 3&4, ‘EF’ (working hydraulics??), ‘pp’ or ‘p’ in figure 7. The same in figure 7 with ‘MSW’, ‘ASW’ and ‘NSW’.

-        [Bar] in figure 6 should be [bar], no capital letter.

-        Figure 11 is not cited in the main text; it seems spelling mistake with figure 12 in page 9 line 217.

-        I recommend that you include the steering concept (a, b or c) in the figure caption of figures 8-12, to help the reader.

TYPOS AND EDITING RECOMMENDATIONS

-        Avoid the use of abbreviations in the abstract, line 1 (it seems A is forgotten in DPS)

-        In line 5 of the abstract, better to use hydraulic than “hyd.”

-        Why using capital letters in line 14 of Introduction for hydraulic power steering

-        It seems that references’ format is not homogenous, since there are titles in capital letters and not. For instance, reference 4 misses the journal name. Please, check it.

-        In the last page, it seems authors forgot to edit the “Abbreviations” section, and it is kept as the template. Please, revise it.

-        Units should not be in italic.

-        From the acknowledgments section, it seems that the prototype building, lab facilities and experimental setup was carried out by Danfoss Power Solutions. As the affiliation of the first author is from Danfoss, can it be concluded that the first author carried out the experimental work and no other company workers?

RECOMENDATION

In general, in the opinion of the reviewer, this paper is a very well-written paper. The reviewer thinks that the topic is of interest and will provide useful information to those who are in the same field as the authors. The research presented is complex and well addressed, the results are satisfactory and the future work is promising. Results are a good resume of the quality of the experimental bench and methodology presented in the paper.

As a weak point, this reviewer would point out the understanding of this new asymmetric concept, the gain and comprehension, and the exact rationale behind of it.

The paper is acceptable for publication with major revision needed; the authors are renowned researchers and experts in the field. However, the reviewer believes that the resulting version of the paper attending these comments will gain in understanding and value.

Author Response

Hi, 

Thank you for the review comments, I have in the best possible way tried to improve the content of the article such that the recommendations are met. 

Best regards Emil Olesen

Reviewer 2 Report

Comments and Suggestions for Authors

enclosed document pdf

Author Response

(The authors gave the same response as above.)

Round 2

Reviewer 1 Report

Comments and Suggestions for Authors

Thanks to the authors for the responses. The paper has gained in value and several parts of it have been improved. The authors have properly answered the reviewer’s comments.

This reviewer acknowledges the effort made by the authors and thanks them for the specific answer to the question related to the more-in-detail description and advantages of an asymmetrical hydraulic steering unit.

As a result, I can accept this paper for publication in its current format.

Reviewer 2 Report

Comments and Suggestions for Authors

No comments